# JAKET: JOINT PRE-TRAINING OF KNOWLEDGE GRAPH AND LANGUAGE UNDERSTANDING

## ABSTRACT

Knowledge graphs (KGs) contain rich information about world knowledge, entities, and relations. Thus, they can be great supplements to existing pre-trained language models. However, it remains a challenge to efficiently integrate information from KG into language modeling. And the understanding of a knowledge graph requires related context. We propose a novel joint pre-training framework, JAKET, to model both the knowledge graph and language. The knowledge module and language module provide essential information to mutually assist each other: the knowledge module produces embeddings for entities in text while the language module generates context-aware initial embeddings for entities and relations in the graph. Our design enables the pre-trained model to easily adapt to unseen knowledge graphs in new domains. Experimental results on several knowledge-aware NLP tasks show that our proposed framework achieves superior performance by effectively leveraging knowledge in language understanding.

## 1 INTRODUCTION

Pre-trained language models (PLM) leverage large-scale unlabeled corpora to conduct self-supervised training. They have achieved remarkable performance in various NLP tasks, exemplified by BERT (Devlin et al., 2018), RoBERTa (Liu et al., 2019b), XLNet (Yang et al., 2019), and GPT series (Radford et al., 2018; 2019; Brown et al., 2020). It has been shown that PLMs can effectively characterize linguistic patterns in text and generate high-quality context-aware representations (Liu et al., 2019a). However, these models struggle to grasp world knowledge about entities and relations (Poerner et al., 2019; Talmor et al., 2019), which are very important in language understanding.

Knowledge graphs (KGs) represent entities and relations in a structural way. They can also solve the sparsity problem in text modeling. For instance, a language model may require tens of instances of the phrase "labrador is a kind of dog" in its training corpus before it implicitly learns this fact. In comparison, a knowledge graph can use two entity nodes "labrador", "dog" and a relation edge "is_a" between these nodes to precisely represent this fact.

Recently, some efforts have been made to integrate knowledge graphs into PLM. Most of them combine the token representations in PLM with representations of aligned KG entities. The entity embeddings in those methods are either pre-computed based on an external source by a separate model (Zhang et al., 2019; Peters et al., 2019), which may not be easily aligned with the language representation space, or directly learned as model parameters (Févry et al., 2020; Verga et al., 2020), which often have an over-parameterization issue due to the large number of entities. Moreover, all the previous works share a common challenge: when the pre-trained model is fine-tuned in a new domain with a previously unseen knowledge graph, it struggles to adapt to the new entities, relations and structure.

Therefore, we propose JAKET, a Joint pre-trAining framework for KnowledgE graph and Text. Our framework contains a knowledge module and a language module, which mutually assist each other by providing required information to achieve more effective semantic analysis. The knowledge module leverages a graph attention network (Veličković et al., 2017) to provide structure-aware entity embeddings for language modeling. And the language module produces contextual representations as initial embeddings for KG entities and relations given their descriptive text. Thus, in both modules, content understanding is based on related knowledge and rich context. On one hand, the joint pre-training effectively projects entities/relations and text into a shared semantic latent space,

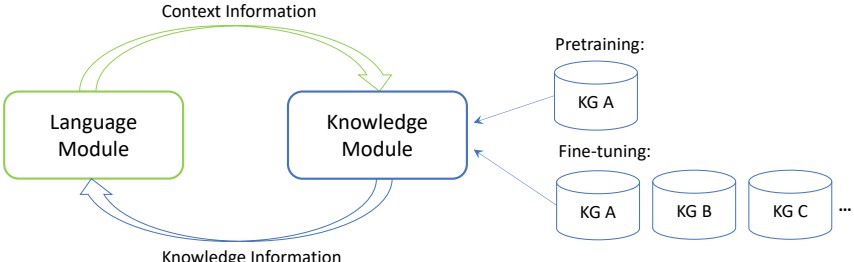

Figure 1: A simple illustration on the novelty of our proposed model JAKET.

which eases the semantic matching between them. On the other hand, as the knowledge module produces representations from descriptive text, it solves the over-parameterization issue since entity embeddings are no longer part of the model's parameters.

In order to solve the cyclic dependency between the two modules, we propose a novel two-step language module $LM_1$ and $LM_2$, respectively. $LM_1$ provides embeddings for both $LM_2$ and KG. The entity embeddings from KG are also fed into $LM_2$, which produces the final representation. $LM_1$ and $LM_2$ can be easily established as the first several transformer layers and the rest layers of a pre-trained language model such as BERT and RoBERTa. Furthermore, we design an entity context embedding memory with periodic update which speeds up the pre-training by 15x.

The pre-training tasks are all self-supervised, including entity category classification and relation type prediction for the knowledge module, and masked token prediction and masked entity prediction for the language module.

A great benefit of our framework is that it can easily adapt to unseen knowledge graphs in the fine-tuning phase. As the initial embeddings of entities and relations come from their descriptive text, JAKET is not confined to any fixed KG. With the learned ability to integrate structural information during pre-training, the framework is extensible to novel knowledge graphs with previously unseen entities and relations, as illustrated in Figure 1.

We conduct empirical studies on several knowledge-aware natural language understanding (NLU) tasks, including few-shot relation classification, question answering and entity classification. The results show that JAKET achieves the best performance compared with strong baseline methods on all the tasks, including those with a previously unseen knowledge graph.

## 2 RELATED WORK

Pre-trained language models have been shown to be very effective in various NLP tasks, including ELMo (Peters et al., 2018), GPT (Radford et al., 2018), BERT (Devlin et al., 2018), RoBERTa (Liu et al., 2019b) and XLNet (Yang et al., 2019). Built upon large-scale corpora, these pretrained models learn effective representations for various semantic structures and linguistic relationships. They are trained on self-supervised tasks like masked language modeling and next sentence prediction.

Recently, a lot of efforts have been made on investigating how to integrate knowledge into PLMs (Levine et al., 2019; Soares et al., 2019; Liu et al., 2020; Guu et al., 2020). These approaches can be grouped into two categories:

1. Explicitly injecting entity representation into the language model, where the representations are either pre-computed from external sources (Zhang et al., 2019; Peters et al., 2019) or directly learned as model parameters (Févry et al., 2020; Verga et al., 2020). For example, ERNIE (THU) (Zhang et al., 2019) pre-trains the entity embeddings on a knowledge graph using TransE (Bordes et al., 2013), while EAE (Févry et al., 2020) learns the representation from pre-training objectives with all the other model parameters. K-BERT (Liu et al., 2020) represents the entities by the embeddings of surface form tokens (i.e. entity names), which contains much less semantic information compared with description text. Moreover, it only injects KG during fine-tuning phase instead of joint-pretraining KG and text.

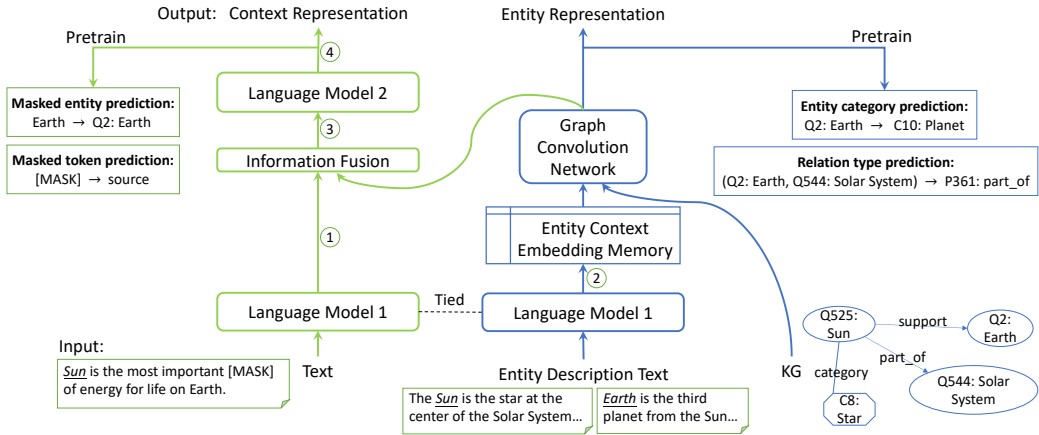

Figure 2: A demonstration for the structure of JAKET, where the language module is on the left side marked green while the knowledge module is on the right side marked blue. Symbol ⊗ indicates the steps to compute context representations introduced in Section 3.4. "QX", "PX" and "CX" are the indices for entities, relations and categories in KG respectively. Entity mentions in text are underlined and italicized such as _Sun_.

2. Implicitly modeling knowledge information, including entity-level masked language modeling (Sun et al., 2019b; Shen et al., 2020), entity-based replacement prediction (Xiong et al., 2019) and knowledge embedding loss as regularization (Wang et al., 2019b). For example, besides token-level masked language modeling, ERNIE (Baidu) (Sun et al., 2019b) uses phrase-level and entity-level masking to predict all the masked slots. KEPLER (Wang et al., 2019b) calculates entity embeddings using a pre-trained language model based on the description text, which is similar to our work. However, they use the entity embeddings for the knowledge graph completion task instead of injecting them into the language model.

Some works (Ding et al., 2019; Lv et al., 2020) investigated the combination of GNN and PLM. For example, Lv et al. (2020) uses XLNet to generate initial node representation based on node context and feeds them into a GNN. However, these approaches do not integrate knowledge into language modeling, and they are designed for specific NLP tasks such as reading comprehension or common-sense reasoning. In comparison, we jointly pre-train both the knowledge graph representation and language modeling and target for general knowledge-aware NLU tasks.

# 3 METHOD

In this section, we introduce the JAKET framework of joint pre-training knowledge graph and language understanding. We begin by defining the mathematical notations, and then present our model architecture with the knowledge module and language module. Finally, we introduce how to pre-train our model and fine-tune it for downstream tasks. The framework is illustrated in Figure 2.

## 3.1 DEFINITION

A knowledge graph is denoted by $\mathcal{KG} = (\mathcal{E}, \mathcal{R}, \mathcal{T})$, where $\mathcal{E} = \{e_1 \dots e_N\}$ is the set of entities and $\mathcal{R} = \{r_1 \dots r_P\}$ is the set of relations. $\mathcal{T} = \{(e_{t_i^1}, r_{t_i^2}, e_{t_i^3}) | 1 \le i \le T, e_{t_i^1}, e_{t_i^3} \in \mathcal{E}, r_{t_i^2} \in \mathcal{R}\}$ stands for the set of head-relation-tail triplets. $N_v = \{(r, u) | (v, r, u) \in \mathcal{T}\}$ represents the set of neighboring relations and entities of an entity $v$.

We define $\mathcal{V} = \{[\text{MASK}], [\text{CLS}], [\text{EOS}], w_1 \dots w_V\}$ as a vocabulary of tokens and the contextual text $\mathbf{x} = [x_1, x_2, \dots, x_L]$ as a sequence of tokens where $x_i \in \mathcal{V}$. In the vocabulary, [MASK] is the special token for masked language modeling (Devlin et al., 2018) and [CLS], [EOS] are the special tokens indicating the beginning and end of the sequence. We define $F$ as the dimension of token embeddings, which is equal to the dimension of entity/relation embeddings from the KG.

The text $\mathbf{x}$ has a list of entity mentions $\mathbf{m} = [m_1, \dots, m_M]$, where each mention $m_i = (e_{m_i}, s_{m_i}, o_{m_i})$: $e_{m_i}$ is the corresponding entity and $s_{m_i}, o_{m_i}$ are the start and end index of this

mention in the context. In other words, $[x_{s_{m_i}}, \ldots, x_{o_{m_i}}]$ is linked with entity $e_{m_i}$[1]. We assume the span of mentions are disjoint for a given text sequence.

As entities in the knowledge graph are represented by nodes without context, we use *entity description text* to describe the concept and meaning of entities. For each entity $e_i$, its description text $\mathbf{x}^{e_i}$ describes this entity. The mention of $e_i$ in $\mathbf{x}^{e_i}$ is denoted as $m^{e_i} = (e_i, s_i^e, o_i^e)$, similarly defined as above. For instance, the description text for the entity "sun" can be "[CLS] The Sun is the star at the center of the Solar System [EOS]". Then the mention is $m^{Sun} = (Sun, 3, 3)$. If there are multiple mentions of $e_i$ in its description text, we choose the first one. If there's no mention of $e_i$ in its description text, we set $s_i^e = o_i^e = 1$. Similarly, we define *relation description text* as the text that can describe each relation.

## 3.2 Knowledge Module

The goal of the knowledge module (KM) is to model the knowledge graph to generate knowledge-based entity representations.

To compute entity node embeddings, we employ the graph attention network (GAT) (Veličković et al., 2017), which uses the self-attention mechanism to specify different weights for different neighboring nodes. However, the vanilla GAT is designed for homogeneous graphs with single-relation edges. To leverage the multi-relational information, we adopt the idea of composition operator (Vashishth et al., 2019) to compose entity embeddings and relation embeddings. In detail, in the $l$-th layer of LM, we update the embedding $E_v^{(l)}$ of entity $v$ as follows:

$$E_v^{(l)} = \text{LayerNorm}\left( \bigoplus_{k=1}^{K} \sigma \left( \sum_{(r,u) \in \mathcal{N}_v} \alpha_{v,r,u}^k W^k f(E_u^{(l-1)}, R_r) \right) + E_v^{(l-1)} \right) \tag{1}$$

$$\alpha_{v,r,u}^k = \frac{\exp\left( \text{LeakyReLU}\left( \mathbf{a}^T \left[ W^k E_v^{(l-1)} \oplus W^k f(E_u^{(l-1)}, R_r) \right] \right) \right)}{\sum_{(r',u') \in \mathcal{N}_v} \exp\left( \text{LeakyReLU}\left( \mathbf{a}^T \left[ W^k E_u^{(l-1)} \oplus W^k f(E_{u'}^{(l-1)}, R_{r'}) \right] \right) \right)} \tag{2}$$

where LayerNorm stands for layer normalization (Ba et al., 2016). $\bigoplus$ means concatenation and $K$ is the number of attention heads. $W^k$ is the model parameter and $R_r$ is the embedding of relation $r$. Note that the relation embeddings are shared across different layers. The function $f(\cdot, \cdot) : \mathbb{R}^F \times \mathbb{R}^F \to \mathbb{R}^F$ merges a pair of entity and relation embeddings into one representation. Here, we set $f(x, y) = x + y$ inspired by TransE (Bordes et al., 2013). More complicated functions like MLP network can also be applied.

The initial entity embeddings $E^{(0)}$ and relation embeddings $R$ are generated from our language module, which will be introduced in Section 3.4. Then, the output entity embeddings from the last GAT layer are used as the final entity representations $E^{\text{KM}}$. Note that the knowledge graph can be very large, making the embedding update over all the entities in Equation (1) not tractable. Thus we follow the minibatch setting (Hamilton et al., 2017): given a set of input entities, we perform neighborhood sampling to generate their multi-hop neighbor sets and we compute representations only on the entities and relations that are necessary for the embedding update.

## 3.3 Language Module

The goal of the language module (LM) is to model text data and learn context-aware representations. The language module can be any model for language understanding, e.g. BERT (Devlin et al., 2018). In this work, we use the pre-trained model RoBERTa-base (Liu et al., 2019b) as the language module.

## 3.4 Solving the cyclic dependency

In our framework, the knowledge and language modules mutually benefit each other: the language module LM outputs context-aware embedding to initialize the embeddings of entities and relations in the knowledge graph given the description text; the knowledge module (KM) outputs knowledge-based entity embeddings for the language module.

---

[1]We do not consider discontinous entity mentions in this work.

However, there exists a cyclic dependency which prevents computation and optimization in this design. To solve this problem, we propose a decomposed language module which includes two language models: $LM_1$ and $LM_2$. We employ the first 6 layers of RoBERTa as $LM_1$ and the remaining 6 layers as $LM_2$. The computation proceeds as follows:

1. $LM_1$ operates on the input text $\mathbf{x}$ and generates contextual embeddings $Z$.

2. $LM_1$ generates initial entity and relation embeddings for KM given description text.

3. KM produces its output entity embeddings to be combined with $Z$ and sent into $LM_2$.

4. $LM_2$ produces the final embeddings of $\mathbf{x}$, which includes both contextual and knowledge information.

In detail, in step 1, suppose the context $\mathbf{x}$ is embedded as $X^{embed}$. $LM_1$ takes $X^{embed}$ as input and outputs hidden representations $Z = LM_1(X^{embed})$.

In step 2, suppose $\mathbf{x}^{e_j}$ is the *entity description text* for entity $e_j$, and the corresponding mention is $m^{e_j} = (e_j, s_j^e, o_j^e)$. $LM_1$ takes the embedding of $\mathbf{x}^{e_j}$ and produces the contextual embedding $Z^{e_j}$. Then, the average of embeddings at position $s_j^e$ and $o_j^e$ is used as the initial entity embedding of $e_j$, i.e. $E_j^{(0)} = (Z_{s_j^e}^{e_j} + Z_{o_j^e}^{e_j})/2$. The knowledge graph relation embeddings $R$ are generated in a similar way using its description text.

In step 3, KM computes the final entity embeddings $E^{KM}$, which is then combined with the output $Z$ from $LM_1$. In detail, suppose the mentions in $\mathbf{x}$ are $\mathbf{m} = [m_1, \ldots, m_M]$. $Z$ and $E^{KM}$ are combined at positions of mentions:

$$Z_k^{merge} = \begin{cases} Z_k + E_{e_{m_i}}^{KM} & \text{if } \exists i \text{ s.t. } s_{m_i} \leq k \leq o_{m_i} \\ Z_k & \text{otherwise} \end{cases} \tag{3}$$

where $E_{e_{m_i}}^{KM}$ is the output embedding of entity $e_{m_i}$ from KM. Then we apply layer normalization (Ba et al., 2016) on $Z^{merge}$: $Z' = \text{LayerNorm}(Z^{merge})$. Finally, $Z'$ is fed into $LM_2$.

In step 4, $LM_2$ operates on the input $Z'$ and obtains the final embeddings $Z^{LM} = LM_2(Z')$. The four steps are marked by the symbol $\otimes$ in Figure 2 for better illustration.

## 3.5 ENTITY CONTEXT EMBEDDING MEMORY

Many knowledge graphs contain a large number of entities. Thus, even for one sentence, the number of entities plus their multi-hop neighbors can grow exponentially with the number of layers in the graph neural network. As a result, it's very time-consuming for the language module to compute context embeddings based on the description text of all involved entities in a batch on the fly.

To solve this problem, we construct an entity context embedding memory, $E^{context}$, to store the initial embeddings of all KG entities. Firstly, the language module pre-computes the context embeddings for all entities and places them into the memory. The knowledge module only needs to retrieve required embeddings from the memory instead of computing them, i.e. $E^{(0)} \leftarrow E^{context}$.

However, as embeddings in the memory are computed from the "old" (initial) language module while the token embeddings during training are computed from the updated language module, there will be an undesired discrepancy. Thus, we propose to update the whole embedding memory $E^{context}$ with the current language module every $T(i)$ steps, where $i$ is the number of times that the memory has been updated (starting from 0). $T(i)$ is set as follows:

$$T(i) = \min(I_{init} * a^{\lfloor i/r \rfloor}, I_{max}) \tag{4}$$

where $I_{init}$ is the initial number of steps before the first update and $a$ is the increasing ratio of updating intervals. $r$ is the number of repeated times of the current updating interval. $I_{max}$ is the maximum number of steps between updates. $\lfloor \cdot \rfloor$ means the operation of rounding down. In our experiments, we set $I_{init} = 10, a = 2, r = 3, I_{max} = 500$, and the corresponding sequence of $T$ is $[10, 10, 10, 20, 20, 20, 40, 40, 40, \ldots, 500, 500]$. Note that we choose $a > 1$ because the model parameters usually change less as training proceeds.

Moreover, we propose a momentum update to make $E^{context}$ evolve more smoothly. Suppose the newly calculated embedding memory by LM is $E^{context}_{new}$, then the updating rule is:

$$E^{context} \leftarrow mE^{context} + (1-m)E^{context}_{new}, \tag{5}$$

where $m \in [0,1)$ is a momentum coefficient which is set as $0.8$ in experiment.

This memory design speeds up our model by about 15x during pre-training while keeping the effectiveness of entity context embeddings. For consideration of efficiency, we use relation embeddings only during fine-tuning.

### 3.6 PRE-TRAINING

During pre-training, both the knowledge module and language module are optimized based on several self-supervised learning tasks listed below. The examples of all the training tasks are shown in Figure 2.

At each pre-training step, we first sample a batch of root entities and perform random-walk sampling on each root entity. The sampled entities are fed into KM for the following two tasks.

**Entity category prediction.** The knowledge module is trained to predict the category label of entities based on the output entity embeddings $E^{\text{KM}}$. The loss function is cross-entropy for multi-class classification, denoted as $\mathcal{L}_c$.

**Relation type prediction.** KM is also trained to predict the relation type between a given entity pair based on $E^{\text{KM}}$. The loss function is cross-entropy for multi-class classification, denoted as $\mathcal{L}_r$.

Then, we uniformly sample a batch of text sequences and their entities for the following two tasks.

**Masked token prediction.** Similar to BERT, We randomly mask tokens in the sequence and predict the original tokens based on the output $Z^{\text{LM}}$ of the language module. We denote the loss as $\mathcal{L}_t$.

**Masked entity prediction.** The language module is also trained to predict the corresponding entity of a given mention. For the input text, we randomly remove $15\%$ of the mentions $\mathbf{m}$. Then for each removed mention $m_r = (e_r, s_r, o_r)$, the model predicts the masked entity $e_r$ based on the mention's embedding. In detail, it predicts the entity whose embedding in $E^{context}$ is closest to $q = g((Z^{\text{LM}}_{s_r} + Z^{\text{LM}}_{o_r})/2)$, where $g(x) = \text{GELU}(xW_1)W_2$ is a transformation function. GELU is an activation function proposed by Hendrycks & Gimpel (2016). Since the number of entities can be very large, we use $e_r$'s neighbours and other randomly sampled entities as negative samples. The loss function $\mathcal{L}_e$ is cross entropy based on the inner product between $q$ and each candidate entity's embedding. Figure 2 shows an concrete example, where the mention "Earth" is not marked in the input text since it's masked and the task is to link the mention "Earth" to entity "Q2: Earth".

### 3.7 FINE-TUNING

During fine-tuning, our model supports using either the knowledge graph employed during pre-training or a novel custom knowledge graph with previously unseen entities[2]. If a custom KG is used, the entity context embedding memory is recomputed by the pre-trained language module using the new entity description text. In this work, we do not update the entity context memory during fine-tuning for consideration of efficiency. We also compute the relation context embedding memory using the pre-trained language model.

## 4 EXPERIMENT

### 4.1 BASIC SETTINGS

**Data for Pre-training.** We use the English Wikipedia as the text corpus, Wikidata (Vrandečić & Krötzsch, 2014) as the knowledge graph, and SLING (Ringgaard et al., 2017) to identify entity mentions. For each entity, we use the first 64 consecutive tokens of its Wikipedia page as its description text and we filter out entities without a corresponding Wikipedia page. We also remove entities that

---

[2]We assume the custom domain comes with NER and entity linking tools which can annotate entity mentions in text. The training of these systems is beyond the scope of this work.

| Model | 5-way 1-shot | 5-way 5-shot | 10-way 1-shot |
|---|---|---|---|
| PAIR (BERT)* | 85.7 | 89.5 | 76.8 |
| PAIR (RoBERTa) | 86.4 | 90.3 | 77.3 |
| PAIR (RoBERTa+GNN) | 86.3 | - | - |
| PAIR (RoBERTa+GNN+M) | 86.9 | - | - |
| PAIR (KnowBERT) | 86.2 | 90.3 | 77.0 |
| PAIR (JAKET) | **87.4** | **92.1** | **78.9** |

Table 1: Accuracy results on the dev set of FewRel 1.0. ⋆ indicates the results are taken from Gao et al. (2019). PAIR is the framework proposed by Gao et al. (2019).

have fewer than 5 neighbors in the Wikidata KG and fewer than 5 mentions in the Wikipedia corpus. The final knowledge graph contains 3,657,658 entities, 799 relations and 20,113,978 triplets. We use the *instance of* relation to find the category of each entity. In total, 3,039,909 entities have category labels of 19,901 types. The text corpus contains about 4 billion tokens.

**Implementation Details.** We initialize the language module with the pre-trained RoBERTa-base (Liu et al., 2019b) model. The knowledge module is initialized randomly. Our implementation is based on the HuggingFace framework (Wolf et al., 2019) and DGL (Wang et al., 2019a). For the knowledge module, we use a 2-layer graph neural network, which aggregates 2-hop neighbors. The number of sampled neighbors in each hop is 10. More details are presented in the Appendix.

**Baselines.** We compare our proposed model JAKET with the pre-trained RoBERTa-base (Liu et al., 2019b) and two variants of our model: RoBERTa+GNN and RoBERTa+GNN+M. The two models have the same model structure as JAKET, but they are not pre-trained on our data. Moreover, the entity and relation context embedding memories of RoBERTa+GNN are randomly generated while the memories of RoBERTa+GNN+M are computed by RoBERTa.

### 4.2 DOWNSTREAM TASKS

**Few-shot Relation Classification**. Relation classification requires the model to predict the relation between two entities in text. Few-shot relation classification takes the $N$-way $K$-shot setting. Relations in the test set are not seen in the training set. For each query instance, $N$ relations with $K$ supporting examples for each relation are given. The model is required to classify the instance into one of the $N$ relations based on the $N \times K$ samples. In this paper we evaluate our model on FewRel (Han et al., 2018), which is a widely used benchmark dataset for few-shot relation classification, containing 100 relations and 70,000 instances.

We use the pre-trained knowledge graph for FewRel as it comes with entity mentions from Wikidata knowledge graph. To predict the relation label, we build a sequence classification layer on top of the output of LM. More specifically, we use the PAIR framework proposed by Gao et al. (2019), which pairs each query instance with all the supporting instances, concatenate each pair as one sequence, and send the concatenated sequence to our sequence classification model to get the score of the two instances expressing the same relation. We do not use relation embeddings in this task to avoid information leakage.

As shown in Table 1, our model achieves the best results in all three few-shot settings. Comparing the results between RoBERTa and RoBERTa+GNN, we see that adding GNN with randomly generated entity features does not improve the performance. The difference between RoBERTa+GNN+M and RoBERTa+GNN demonstrates the importance of generating context embedding memory by the language module, while JAKET can further improve the performance by pre-training. We also compare with a strong knowledge-enhanced PLM KnowBERT (Peters et al., 2019), which is also pretrained on English Wikipedia and Wikidata KG. The results show that JAKET consistently outperform KnowBERT in different few-shot settings.

**KGQA**. The Question Answering over KG (KGQA) task is to answer natural language questions related to a knowledge graph. The answer to each question is an entity in the KG. This task requires an understanding over the question and reasoning over multiple entities and relations.

We use the vanilla version of the MetaQA (Zhang et al., 2017) dataset, which contains questions requiring multi-hop reasoning over a novel movie-domain knowledge graph. The KG contains 135k triplets, 43k entities and 9 relations. Each question is provided with one entity mention and the

| Model | KG-Full | | KG-50% | |
|---|---|---|---|---|
| | 1-hop | 2-hop | 1-hop | 2-hop |
| RoBERTa | 90.2 | 70.8 | 61.5 | 39.3 |
| RoB+G+M | 91.4 | 72.6 | 62.5 | 40.8 |
| JAKET | **93.9** | **73.2** | **63.1** | **41.9** |

Table 2: Results on the MetaQA dataset over 1-hop and 2-hop questions under *KG-Full* and *KG-50%* settings. RoB+G+M is the abbreviation for the baseline model RoBERTa+GNN+M.

| Model | 100% | 20% | 5% |
|---|---|---|---|
| GNN | 48.2 | - | - |
| RoBERTa | 33.4 | - | - |
| RoB+G+M | 79.1 | 66.7 | 53.5 |
| JAKET | **81.6** | **70.6** | **58.4** |

Table 3: Results on the entity classification task over an unseen Wikidata knowledge graph. RoB+G+M is the abbreviation for the baseline model RoBERTa+GNN+M.

question is named as a $k$-hop question if the answer entity is a $k$-hop neighbor of the question entity. We define all the $k$-hop neighbor entities of the question entity as the candidate entities for the question. We also consider a more realistic setting where we simulate an incomplete KG by randomly dropping a triplet with a probability $50\%$. This setting is called *KG-50%*, compared with the full KG setting *KG-Full*. For each entity, we randomly sample one question containing it as the entity's description context. We manually write the description for each relation since the number of relations is very small. We use the output embedding of [CLS] token from LM as the question embedding, and then find the entity with the closest context embedding.

As shown in Table 2, RoBERTa+GNN+M outperforms RoBERTa, demonstrating the effectiveness of KM+LM structure. JAKET further improves the accuracy by 0.6% to 2.5% under both KG settings, showing the benefits of our proposed joint pre-training.[3]

**Entity Classification**. To further evaluate our model's capability to reason over unseen knowledge graphs, we design an entity classification task. Here, the model is given a portion of the Wikidata knowledge graph unseen during pre-training, denoted as $\mathcal{KG}'$. It needs to predict the category labels of these novel entities. The entity context embeddings are obtained in the same way as in pre-training. The relation context embeddings are generated by its surface text. The number of entities and relations in the $\mathcal{KG}'$ are 23,046 and 316 respectively. The number of triplets is 38,060. Among them, 16,529 entities have 1,291 distinct category labels. We conduct experiments under a semi-supervised transductive setting by splitting the entities in $\mathcal{KG}'$ into train/dev/test splits of 20%, 20% and 60%. To test the robustness of models to the size of training data, we evaluate models when using 20% and 5% of the original training set.

In this task, RoBERTa takes the entity description text as input for label prediction while neglecting the structure information of KG. JAKET and RoBERTa+GNN+M make predictions based on the entity representation output from the knowledge module. We also include GNN as a baseline, which uses the same GAT-based structure as our knowledge module, but with randomly initialized model parameters and context embedding memory. GNN then employs the final entity representations for entity category prediction.

As shown in Table 3, our model achieves the best performance under all the settings. The performance of GNN or RoBERTa alone is significantly lower than JAKET and RoBERTa+GNN+M, which demonstrates the importance of integrating both context and knowledge information using our proposed framework. Also, the gap between JAKET and RoBERTa+GNN+M increases when there's less training data, showing that the joint pre-training can reduce the model's dependence on downstream training data.

## 5 CONCLUSION

This paper presents a novel framework, JAKET, to jointly pre-train models for knowledge graph and language understanding. Under our framework, the knowledge module and language module both provide essential information for each other. After pre-training, JAKET can quickly adapt to unseen knowledge graphs in new domains. Moreover, we design the entity context embedding memory which speeds up the pre-training by 15x. Experiments show that JAKET outperforms baseline methods in several knowledge-aware NLU tasks: few-shot relation classification, KGQA and entity classification. In the future, we plan to extend our framework to natural language generation tasks.

---

[3]For fair comparison, we do not include models which incorporate a dedicated graph retrieval module (Sun et al., 2018; 2019a)

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

## A  APPENDIX

### A.1  IMPLEMENTATION DETAILS

The dimension of hidden states in the knowledge module is 768, the same as RoBERTa-base, and the number of attention heads is 8. During pre-training, the batch size and length of text sequences are 1024 and 512 respectively. The batch size of KG entities is 16,384. The number of training epochs is 8. JAKET is optimized by AdamW (Loshchilov & Hutter, 2017) using the following parameters: $\beta_1 = 0.9$, $\beta_2 = 0.999$, $\epsilon = $ 1e-8, and weight decay of $0.01$. The learning rate of the language module is warmed up over the first 3,000 steps to a peak value of 1e-5, and then linearly decayed. The learning rate of our knowledge module starts from 1e-4 and then linearly decayed.

### A.2  COMPUTATION ANALYSIS

The computation of the KG module is much less than the LM module. For BERT-base or RoBERTa-base, the number of inference computation flops (#flops) over each sequence (length 128) is over 22 billion [1, 2]. Here, we theoretically compute the number of flops of the KG module as follows: The sequence length $N = 128$, and hidden dimension $H = 768$. The number of entities in a sequence is usually less than $N/5$. The number of sampled neighbors per entity $r = 10$. And the number of layers of the GNN based KG module $L = 2$. It follows that the #flops of KG module is about $N/5 \times r^L \times 2H^2 \approx 3$ billion, less than $1/7$ of LM computation. If we set $r = 5$, the #flops can be further reduced to about $1/30$ of LM computation.

During pre-training, another computation overhead is entity context embedding memory update (Section 3.5): Firstly, the number of entities is about 3 million and the update step interval is 500. Thus for each step on average the model processes the description text of $3e6/500 = 6e3$ entities. Secondly, the length of description text is 64, much smaller than the length of input text 512, and we only use LM1 (the first half of LM module) for entity context embedding generation, which saves half of the computation time compared to using the whole LM module. Thirdly, the embedding update only requires forward propagation, costing only half of computation compared to training process which requires both forward and backward propagation. Thus, generating context embedding of 6k entities consumes about the same number of flops as training $6000 \times 64/(512 \times 2 \times 2) \approx 200$ input texts, much smaller than the batch size 1024. In short, the entity context embedding memory update only costs $200/1024 \approx 1/5$ additional computation. Note this computation overhead only exists during pre-training, since entity embedding memory is not updated when fine-tuning.

