# OpenReview forum: "JAKET: Joint Pre-training of Knowledge Graph and Language Understanding"
_ICLR.cc/2021/Conference — Reject_

### Official Review · AnonReviewer1 · 2020-10-13
**A joint KG and language pre-training model**

**Rating:** 5
**Confidence:** 4

**Review:**

This work proposes a method for joint pre-training of knowledge graph and text data which embeds KG entities and relations into shared latent semantic space as entity embeddings from text. The proposed model JAKET consists of two main parts: a language module and a knowledge module. The model is pre-trained on a collection of tasks: entity category prediction, relation type prediction, masked token prediction and masked entity prediction. The proposed framework enables fine-tuning on knowledge graphs which are unseen during pre-training.


Overall, I believe that the work on knowledge-enhanced language models to be an interesting and important area of research. However, I believe the paper is not ready for publication in its current form as (i) the demonstrated improvements obtained by pre-training with the added KG module seem minor compared to the computational overhead of having to compute entity and relation embeddings using GNNs; and (ii) experimental comparison to some relevant prior work is missing.

Questions/comments for the authors:

1. One of the drawbacks of the proposed method is that it assumes entity descriptions to always be available, which might be the case for Wikidata, but it is not usually the case with e.g. standard knowledge graph completion datasets WN18 and FB15k. How would fine-tuning work on knowledge graphs that do not have entity descriptions?
2. What does M in RoBERTa+GNN+M stand for? Is it memory?
3. The improvements over a pure language model on few-shot relation classification and KGQA are minor, especially given the computational overhead that adding a KG module entails. The authors should include a discussion on computational overhead of having a KG module vs a pure language model.
4. The experimental comparison to existing knowledge-enhanced language models from Section 2 is missing.

---

> ### Author Response · Authors · 2020-11-21
> **Response to Reviewer 1**
>
> Dear Reviewer 1, we really appreciate your valuable feedback.
>
> First, we want to summarize the main contributions of our paper:
>
> 1. We propose a novel knowledge-language co-pretraining framework, JAKET, where a knowledge module and a language module mutually assist each other for more effective semantic analysis.
> 2. We use a two-stage language model to solve the cyclic dependency problem between two modules.
> 3. We employ an entity context embedding memory with scheduled updates which speeds up the pre-training by 15x.
> 4. JAKET can easily adapt to unseen knowledge graphs in the finetuning phase.
> 5. Experiments show that JAKET outperforms strong baseline methods on the tasks of few-shot relation classification, question answering, and entity classification.
>
> Here are the replies to each of your question/comment:
>
> **1. For the question “How would fine-tuning work on knowledge graphs that do not have entity descriptions?”:**
>
> There are multiple ways to obtain entity descriptions: find related entries in Wikidata (https://www.wikidata.org/) or Wiktionary (https://www.wiktionary.org/); retrieve sentences in the corpus containing the entity. In any case, the goal is to obtain better entity representations by placing them into context. Finally, the backup is to use the entity name itself.
>
> **2. For the question “What does M in RoBERTa+GNN+M stand for? Is it memory?”:**
>
> Yes, M stands for Memory. More precisely, it means that the memories for entity and relation context embedding are computed by RoBERTa on their description text. In comparison, the memories of RoBERTa+GNN are randomly initialized.
>
> **3. For the comment on “computational overhead that adding a KG module”:**
>
> Good suggestion! The computation of the KG module is much less than the LM module. For BERT-base or RoBERTa-base, the number of inference computation flops (#flops) over each sequence (length N=128) is over 22 billion [1, 2]. Here, we compute the number of flops of the KG module:
> The sequence length is N=128, and the hidden dimension is H=768. The number of entities in a sequence is usually less than N/5. The number of sampled neighbors per entity is r=10. And the number of layers of the GNN based KG module is L=2.
> It follows that the #flops of KG module is about N/5 \* r^L \* 2H^2 ≈ 3 billion, less than 1/7 of LM computation. If we set r=5, the #flops can be further reduced to about 1/30 of LM computation.
>
> During pre-training, another computation overhead is entity context embedding memory update (Sec. 3.5):
> Firstly, the number of entities is about 3 million and the update step interval is 500. Thus for each step on average the model processes the description text of 3e6/500=6e3 entities. Secondly, the length of description text is 64, much smaller than the length of input text 512, and we only use LM1 (the first half of LM module) for entity context embedding generation, which saves half of the computation time compared to using the whole LM module. Thirdly, the embedding update only requires forward propagation, costing only half of computation compared to training process which requires both forward and backward propagation. Thus, generating context embedding of 6k entities consumes about the same number of flops as training 6000\*64/(512\*2\*2) ≈ 200 input texts, much smaller than the batch size 1024. In short, the entity context embedding memory update only costs 200/1024≈1/5 additional computation. We can also set a larger update step interval to further reduce the computation cost. Note this computation overhead only exists during pre-training, since entity embedding memory is not updated when fine-tuning.
>
> We've included the discussion in our revised paper. For the concern about performance improvement, we think this depends on the tasks: For the tasks that rely much on the semantic information of the input text itself (which can be well captured by pure LM), the improvement can be relatively small [3, 4]. For the tasks which rely much on KG information like semi-supervised entity classification, our model performs much better than the pure language model like RoBERTa.
>
>
> [1] MobileBERT: a Compact Task-Agnostic BERT for Resource-Limited Devices (ACL 2020)
> [2] TinyBERT: Distilling BERT for Natural Language Understanding (EMNLP 2020)
> [3] Knowledge enhanced contextual word representations (EMNLP 2019)
> [4] ERNIE: Enhanced Language Representation with Informative Entities (ACL 2019)

---

> > ### Author Response · Authors · 2020-11-21
> > **Continued response to Reviewer 1**
> >
> > **4. About comparison to existing knowledge-enhanced language models**
> >
> > We conducted an additional experiment on the FewRel dataset using one strong knowledge-enhanced pre-training model KnowBERT [1], which is also pretrained on English Wikipedia corpora and Wikidata KG. We fine-tune the pre-trained model provided in its official codebase. The result below demonstrates the effectiveness of our proposed co-training framework. We’ve included the result in the revised version of our paper.
> >
> > | FewRel   | 5-way-1-shot | 5-way-5-shot | 10-way-1-shot |
> > |----------|--------------|--------------|---------------|
> > | KnowBERT |     86.2     |     90.3     |      77.0     |
> > | JAKET    |     87.4     |     92.1     |      78.9     |
> >
> > [1] Knowledge enhanced contextual word representations (EMNLP 2019)

---

> > > ### Comment · AnonReviewer1 · 2020-11-25
> > > **Response after rebuttal**
> > >
> > > Thank you for taking the time to respond to my questions and updating the paper. I'm no longer concerned with the computational overhead, however, I still find the empirical results unconvincing. As other reviewers have also pointed out, I believe the proposed model should be compared to other existing baselines (e.g. KnowBERT, K-BERT) on all the proposed tasks (not just few-shot relation classification which you've included in the revised version, but also KGQA and entity classification). Thus, I've decided to keep my initial rating.

---

### Official Review · AnonReviewer3 · 2020-10-28
**JAKET Review**

**Rating:** 5
**Confidence:** 5

**Review:**

The paper proposed a pretraining framework for language models and knowledge graphs. As authors mentioned in the paper, many approaches recently focused on this topic, with different variations with respect to entity embedding, initialization, over-parametrization, masked language model, fine-tuning task and etc.

Novelty: The authors mentioned entity embeddings from previous works in this area mainly are computed by external resources and are injected  to the model or they are learned as parameters of the model. What about K-BERT? Doesn’t seem to have these problems. The paper doesn’t provide a comprehensive comparison with K-Bert. I think the novelty of the work may need to be further justified.
Also, what is the relation text or description? What kind of description do the relations have? Is it useful to use Roberta for relation embedding initialization?

Experiments:
The paper proposes a few baselines and compares the architecture with those baselines. However, there are several other works in this area. Why not comparing with those related approaches?  As an example comparison with K-BERT?

It would be also interesting to have more baselines without language models (like the GNN in Entity Classification task)? For example, just initialize the knowledge graph by a pretrained language model and evaluate the knowledge graph on questions answering? Or even have a GNN on top of that without language model? Comparison with other knowledge graph embedding approaches on Wikipedia? Also, what if we separately pretrain knowledge graph and language model. How does affect the experiments?

There is a lack of experiments to show the Effect of number of steps in graph embedding update and also training time?

---

> ### Author Response · Authors · 2020-11-21
> **Response to Reviewer 3**
>
> Dear Reviewer 3, we really appreciate your valuable feedback.
>
> First, we want to summarize the main contributions of our paper:
>
> 1. We propose a novel knowledge-language co-pretraining framework, JAKET, where a knowledge module and a language module mutually assist each other for more effective semantic analysis.
> 2. We use a two-stage language model to solve the cyclic dependency problem between two modules.
> 3. We employ an entity context embedding memory with scheduled updates which speeds up the pre-training by 15x.
> 4. JAKET can easily adapt to unseen knowledge graphs in the finetuning phase.
> 5. Experiments show that JAKET outperforms strong baseline methods on the tasks of few-shot relation classification, question answering, and entity classification.
>
> Here are the replies to each of your question/comment:
>
> **1. For the question about the difference between our model JAKET and K-BERT:**
>
> Thanks for bringing this up. There are two main differences between JAKET and K-BERT: 1. K-BERT only injects KG triplets during fine-tuning, while JAKET jointly learns embeddings for entities and tokens during both pre-training and fine-tuning. 2. K-BERT represents the entities and relations by surface form tokens (i.e. names), while JAKET utilizes their description texts, which contain much more semantic information. We’ve made this clear in the revised paper.
>
> **2. For the question “What is the relation text or description”:**
>
> The relation description is similar to entity description, which is a piece of text describing the relation. For example, the relation “instance of” in Wikidata has a provided description “that class of which this subject is a particular example and member”. This will provide richer information about the relation beyond its surface name. Experiments show that using LM such as RoBERTa to initialize relation embedding using the description text can improve the performance on downstream tasks like MetaQA by 2~3%.
>
> **3. About comparison with other knowledge pretrained LM models.**
>
> We conducted an additional experiment on the FewRel dataset using one strong knowledge-enhanced pre-training model KnowBERT [1], which is also pretrained on English Wikipedia corpora and Wikidata KG. We fine-tune the pre-trained model provided in its official codebase. The result below demonstrates the effectiveness of our proposed co-training framework. We’ve included the result in the revised version of our paper.
>
> | FewRel   | 5-way-1-shot | 5-way-5-shot | 10-way-1-shot |
> |----------|--------------|--------------|---------------|
> | KnowBERT |     86.2     |     90.3     |      77.0     |
> | JAKET    |     87.4     |     92.1     |      78.9     |
>
> **4. For the suggestions about more baselines:**
>
> Thanks for suggesting these reasonable baselines. For the KGQA task, it usually requires a pipeline of methods including question embedding, sub-graph retrieval, graph modeling, answer selection, etc. We test our model’s capability on the subtask of generating question representation. Thus we only compare with models that focus on sentence representation generation like RoBERTa. For the comparison with other KG embedding methods and a separate pre-training framework, we’re unable to provide ablation results due to the limited time of rebuttal. But we’ll include them in the final version of our paper.
>
> **5. For the question about “Effect of number of steps in graph embedding update and also training time”**
>
> We tried different updating intervals and found the current schedule can achieve the lowest and most stable pre-training loss.
> The update of entity context embedding memory takes only 1/6 of the pre-training time. We explain that in the following calculation:
> Firstly, the number of entities is about 3 million and the update step interval is 500. Thus for each step on average, the model processes the description text of 3e6/500=6e3 entities. Secondly, the length of the description text is 64, much smaller than the length of input text 512, and we only use LM1 (the first half of LM module) for entity context embedding generation, which saves half of the computation time compared to using the whole LM module. Thirdly, the embedding update only requires forward propagation, costing only half of computation compared to training process which requires both forward and backward propagation. Thus, generating context embedding of 6k entities consumes about the same number of flops as training 6000\*64/(512\*2\*2) ≈ 200 input texts, much smaller than the batch size 1024. In short, the entity context embedding memory update only costs 200/1024≈1/5 additional computation. Note this computation overhead only exists during pre-training, since entity embedding memory is not updated when fine-tuning.
>
> [1] Knowledge enhanced contextual word representations (EMNLP 2019)

---

### Official Review · AnonReviewer4 · 2020-10-30
**Review for JAKET**

**Rating:** 6
**Confidence:** 4

**Review:**

# Summary

This paper proposed a new language modeling pretraining method that leverages the knowledge graph information. Specifically, the paper replaces the entity embedding in one hidden layer of BERT context embedding, with the corresponding graph attention embedding that is obtained from the knowledge graph. The pretraining tasks contain not only the language related tasks (like predicting masked tokens), but also the knowledge graph tasks like entity classification or relation type prediction. Experiments on few-shot learning tasks, question answering and entity classification show better performance over other pretraining counterparts.


# Pros

- The motivation of combining knowledge graphs and unstructured text data for pretraining is valid.
- The joint training architecture is interesting.
- The improvement over other kg+lm or purely lm baselines is consistent.


# Cons

- Some of the designs need justification.
- Some of the baselines/ablations might be missing.
- The experiments have some flaws.


# Details

Overall I lean towards accepting the paper, if my concerns can be properly addressed.

I like the idea of having a joint training method for both the kg embedding and text embedding. Although it encounters additional implementation/computation difficulties, it entangles the two sets of information in a coherent way. Also the pretraining has been done on a large scale, with a reasonable size of knowledge graph as the backbone. I think the paper has demonstrated a nontrivial contribution to the field.

There are several potential issues with the current paper:

1. It seems the paper sacrifices too much on the efficiency, while having many heuristic approximations/designs that are un-justified. For example, a) how much would the entity embedding update affect the fine-tuning; b) how does the neighborhood sampling affect the knowledge graph embedding in the context of BERT training;

2. It is not sure how each individual pretraining method affects the quality of the embedding. Also for knowledge graphs, a commonly used embedding learning method is the link prediction, where one samples positive/negative <e1, r, e2> triplets for learning. The entity category prediction or relation type prediction is less common. Would it be helpful to do the link prediction instead?

3. An important baseline is missing, where one first pretrain the KG embedding and BERT embedding separately, and concat/merge them with downstream fine-tuning tasks. I’d like to see the results on the three tasks in the experiments.

4. For table 2 it would be nice to include the knowledge graph based approaches that don’t leverage the pretraining, including the pullnet results, or the results that come with original metaQA paper. I don’t get the comment at the bottom of page 8 why approaches like VRN in MetaQA paper are not ‘fair’, as these models actually use less information.

5. For table 1 and 2, the improvement over RoBERTa seems marginal. However for table 3 the gap is huge. Could the authors provide explanations for
1) why use entity classification tasks? It is somewhat unfair, as the pretraining of the proposed method already involved with the entity classification tasks (although in a transductive setting); Also as I mentioned in 2, link prediction would be a more preferable task for evaluating the quality of the knowledge graph embedding;
and 2) why is the gap so large, is it due to the issue in 1)?

# Questions

I’d like to hear the answers to my questions above.

# Improvement

I highly encourage the authors to include the necessary baselines in experiments, as suggested above.

---

> ### Author Response · Authors · 2020-11-21
> **Response to Reviewer 4**
>
> Dear Reviewer 4, we really appreciate your valuable feedback.
>
> First, we want to summarize the main contributions of our paper:
>
> 1. We propose a novel knowledge-language co-pretraining framework, JAKET, where a knowledge module and a language module mutually assist each other for more effective semantic analysis.
> 2. We use a two-stage language model to solve the cyclic dependency problem between two modules.
> 3. We employ an entity context embedding memory with scheduled updates which speeds up the pre-training by 15x.
> 4. JAKET can easily adapt to unseen knowledge graphs in the finetuning phase.
> 5. Experiments show that JAKET outperforms strong baseline methods on the tasks of few-shot relation classification, question answering, and entity classification.
>
> Here are the replies to each of your question/comment:
>
> **1. For the issue about “un-justified design like entity embedding update and neighbor sampling”**
>
> Thanks for the suggestion. Due to the limited time of rebuttal, we’re unable to rerun pre-training to provide ablation results. But we’ll include them in the final version of our paper.
>
> **2. For the question “for knowledge graphs, ... do the link prediction instead?”**
>
> Link prediction can be easily combined into our framework and is definitely worth trying. For the two pre-training tasks in our paper, entity category prediction has been demonstrated to be effective in pre-training graph neural networks [1]. Relation type prediction can be regarded as one type of link prediction, as it also masks one element (i.e. relation) of a triplet (<e1, [MASK], e2>) and predicts the masked element.
>
> **3. For the question “baseline … pretrain the KG embedding and BERT embedding separately, and concat/merge them ...”.**
>
> This is a good baseline which we’ll include in the final paper. We add a comparison with one strong knowledge-enhanced PLM KnowBERT [2], which pre-computes the KG embedding and combines them with token embeddings in BERT instead of jointly training them. The result of the FewRel dataset shows that our model outperforms KnowBERT. This further demonstrates the reason we combine the pre-training of knowledge graph and language modeling: i) the knowledge graph can provide information about related entities and relations for entities in the text; ii) the language model can provide contextual information of entity nodes/relation edges in the knowledge graph based on description text. In this way, both modules can benefit from each other by leveraging richer information.
>
> | FewRel   | 5-way-1-shot | 5-way-5-shot | 10-way-1-shot |
> |----------|--------------|--------------|---------------|
> | KnowBERT |     86.2     |     90.3     |      77.0     |
> | JAKET    |     87.4     |     92.1     |      78.9     |
>
> **4. For question about “why approaches like VRN in MetaQA paper are not ‘fair’”**
>
> VRN or PullNet is a pipeline specifically designed for Question answering over KG, including question embedding, sub-graph retrieval, graph modeling, answer selection, etc. In comparison, our model is a generalized framework for natural language understanding. In the case of MetaQA, we test our model’s capability on the subtask of generating question representation. Thus we only compare with models that focus on sentence representation generation like RoBERTa.
>
> **5. For the question “why use entity classification tasks? ... Why is the gap so large?”**
>
> First, we want to clarify that the task is under fair comparison since the downstream Wikidata KG is disjoint with the pre-training Wikidata KG, i.e. both the entities and labels are not seen during pre-training. Second, semi-supervised node classification is also an important NLP task and has been intensively studied recently [3, 4]. The gap is large because this task relies more on the structure information of KG, which can not be captured by vanilla language models like RoBERTa. Similar gap has also been shown in [3, 4].
>
> [1] Strategies for Pre-training Graph Neural Networks (ICLR 2020)
> [2] Knowledge enhanced contextual word representations (EMNLP 2019)
> [3] Semi-Supervised Classification with Graph Convolutional Networks (ICLR 2016)
> [4] Modeling Relational Data with Graph Convolutional Networks (ESWC 2018)

---

### Official Review · AnonReviewer2 · 2020-10-31
**An interesting paper**

**Rating:** 5
**Confidence:** 5

**Review:**

This paper presents an approach to jointly pre-train language models and representations for knowledge graphs. In particular, natural language texts (English Wikipedia) are used to train context representations, while knowledge graphs (Wikidata) train entity representations (and both depend on each other). Experiments show that the approach outperforms baseline methods on several natural language understanding tasks: few-shot relation classification, knowledge graph question answering, and entity classification.

The presented approach looks very promising, however, it also leaves several doubts. One natural ablation question is whether we could simply include the knowledge graph as plain text in training only a language model (via a similar pre-training and fine-tuning)? So, whether the additional hybrid structure involving graph convolution networks is actually necessary? (Maybe the gain in accuracy is only due to the additional information that is available in the knowledge graph?) Another natural question is why the authors have not made further experiments on other datasets and knowledge graphs? Does this approach only work for English Wikipedia and Wikidata (maybe because these two are matching extraordinarily well)? Finally, I would have expected further experimental comparisons to related approaches.

After rebuttal: I'm also still not convinced by the experimental evaluation. For this reason, I slightly downgraded my overall rating.

---

> ### Author Response · Authors · 2020-11-21
> **Response to Reviewer 2**
>
> Dear Reviewer 2, We really appreciate your valuable feedback.
>
> First, we want to summarize the main contributions of our paper:
> 1. We propose a novel knowledge-language co-pretraining framework, JAKET, where a knowledge module and a language module mutually assist each other for more effective semantic analysis.
> 2. We use a two-stage language model to solve the cyclic dependency problem between two modules.
> 3. We employ an entity context embedding memory with scheduled updates which speeds up the pre-training by 15x.
> 4. JAKET can easily adapt to unseen knowledge graphs in the finetuning phase.
> 5. Experiments show that JAKET outperforms strong baseline methods on the tasks of few-shot relation classification, question answering, and entity classification.
>
> Here are the replies to each of your question/comment:
>
> **1. For the question “... include the knowledge graph as plain text in training only a language model?”**
>
> Converting a knowledge graph to plain text is one way to combine knowledge with language modeling. However, since KGs represent facts in a topological way, conversion to text will inevitably lose this structural information. Thus, we use the graph neural network to model the topological information via neighborhood aggregation and the contextual information via initial node embeddings from description text.
>
> **2. For the question “... not made further experiments on other datasets and knowledge graphs? … only work for English Wikipedia and Wikidata?”:**
>
> In this paper we conducted experiments on three different tasks: few-shot relation classification, question answering over KG and entity classification, with both KG used during pre-training and novel KG with unseen entities and relations. We are also planning to apply JAKET to other tasks and languages in the future. Note that the JAKET architecture we propose is a framework that can adapt any corpus and related knowledge graph. The design of the framework decouples the source of knowledge from the model. This has been shown by replacing Wikidata with movie KG in the experiments on the MetaQA dataset.
>
> **3. For the suggestion about “further experimental comparisons to related approaches”:**
>
> Good suggestion. We conducted an additional experiment on the FewRel dataset using one strong knowledge-enhanced pre-training model KnowBERT [1], which is also pretrained on English Wikipedia corpora and Wikidata KG. We fine-tune the pre-trained model provided in its official codebase. The result below demonstrates the effectiveness of our proposed co-training framework. We’ve included the result in the revised version of our paper.
>
> | FewRel   | 5-way-1-shot | 5-way-5-shot | 10-way-1-shot |
> |----------|--------------|--------------|---------------|
> | KnowBERT |     86.2     |     90.3     |      77.0     |
> | JAKET    |     87.4     |     92.1     |      78.9     |
>
> [1] Knowledge enhanced contextual word representations (EMNLP 2019)

---

### Author Response · Authors · 2020-11-21
**Paper Revision**

We thank the reviewers for their valuable feedback. Based on the questions and suggestions, we made the following content revisions in our paper:

1. Add KnowBERT [1] results on the FewRel dataset in Table 1 on Page 7. Add analysis between KnowBERT and JAKET in the third paragraph of Section 4.2.
2. Add computation analysis in Appendix A.2.
3. Add difference discussion with K-BERT [2] in the third paragraph of Section 2.

[1] Knowledge enhanced contextual word representations (EMNLP 2019)
[2] K-BERT: Enabling Language Representation with Knowledge Graph (AAAI 2020)

---

### Decision · Program_Chairs · 2021-01-07
**Final Decision**

**Decision:**

Reject

**Comment:**

Four knowledgeable referees reviewed this paper; one reviewer (weakly) supports accept and other three indicate reject. Even with the rebuttal, all reviewers (including positive reviewer) have concerns on unconvincing experimental results (due to missing baselines for instance). I basically agree on negative reviews that this submission fails to have enough quality considering the high standard of ICLR.